# Influence of Pasture on Stearoyl-CoA Desaturase and miRNA 103 Expression in Goat Milk: Preliminary Results

**DOI:** 10.3390/ani9090606

**Published:** 2019-08-26

**Authors:** Raffaella Tudisco, Valeria Maria Morittu, Laura Addi, Giuseppe Moniello, Micaela Grossi, Nadia Musco, Raffaella Grazioli, Vincenzo Mastellone, Maria Elena Pero, Pietro Lombardi, Federico Infascelli

**Affiliations:** 1Department of Veterinary Medicine and Animal Production, University of Napoli Federico II, 80137 Napoli, Italy; 2Department of Health Sciences, Magna Graecia University of Catanzaro, 88100 Catanzaro, Italy; 3Department of Veterinary Medicine, University of Sassari, 07100 Sassari, Italy

**Keywords:** goat, miRNA, stearoyl-CoA desaturase, CLA

## Abstract

**Simple Summary:**

An experiment to determine the effect of pasture on stearoyl-CoA desaturase (SCD) and micro-RNA (miRNA) 103 expression was carried out on dairy goats. SCD is involved in determining milk content of conjugated linoleic acids (CLAs) that are considered an important health factor in human nutrition. The alterations of the normal pathway of expression of miRNAs can have consequences on the normal cellular physiology and lead to different types of pathologies. The pasture significantly affected milk fat as well as fatty acid profile in goats, in particular CLAs showed higher levels in grazing animals with potential beneficial effects on human health. The pasture affected only the SCD trend, while that of miRNA 103 was influenced only by the stage of lactation. Due to the increasing interest of consumers for the healthy aspects of foods of animal origin, there is an important ongoing debate in the scientific community concerning those factors affecting milk quality in terms of human health. To our knowledge, this is the first observation of the effects of pasture on miRNA expression in milk from ruminant species.

**Abstract:**

The effect of pasture on the stearoyl-CoA desaturase (SCD) and miRNA 103 expression was evaluated on dairy goats divided into two homogeneous groups (G, grazing, and S, stable). Group S was housed in a stall and received alfalfa hay as forage, while group G was led to pasture. The goats of both the groups received the same amount of concentrate. Milk yield did not differ statistically between the groups. Group G showed significantly higher fat (4.10% vs. 2.94%, *p* < 0.01) and protein percentage (3.43% vs. 3.25%; *p* < 0.05) than group S. Among milk fatty acids, group S showed significantly higher levels of saturated fatty acids (SFA) and lower values of mono-unsaturated fatty acid (MUFA). The percentages of polyunsaturated fatty acid (PUFA) were not different between groups even if pasture significantly affected the percentages of C18:3 and total omega 3. In group G, total CLAs were twice than in group S (0.646% vs. 0.311%; *p* < 0.01) mainly due to the differences in CLA cis9 trans 11 (0.623% vs. 0.304%; *p* < 0.01). Milk total CLA in grazing group was significantly (*p* < 0.01) higher in August according to the highest value of both linoleic and α-linolenic acids in the pasture. In grazing animals, SCD expression decreased from April to June, increased in July and decreased again in August, while it was almost unvaried along the trial in group S. By contrast, the expression of miRNA 103 showed a similar trend for both groups, decreasing from April to June, increasing in July and falling down in August. To our knowledge, this is the first observation of the effects of pasture on miRNA expression in milk from ruminant species.

## 1. Introduction

It is well known that ruminants’ diet greatly influences the milk nutritional characteristics, particularly fatty acid profile [1]. This is of great importance because of the beneficial effects of some long-chain fatty acids for human health. Among the others, the conjugated linoleic acids (CLAs), a group of positional and geometric fatty acid isomers of linoleic acid, whose major isomer (up to 80% of total CLA) is cis-9, trans-11C18:2 or rumenic acid (RU), are considered to have immunomodulating, anti-carcinogenic and anti-artheriosclerosis properties [2,3]. In milk, CLAs come from the activity of rumen bacteria on dietary unsaturated fatty acids but also from the action of mammary gland stearoyl-CoA desaturase (SCD) on trans-11 C18:1 (TVA, trans vaccenic acid), an intermediate product, obtained during polyunsaturated fatty acids biohydrogenation. SCD also acts on the biosynthesis of monounsaturated fatty acids by a cis double bond between carbons 9 and 10 in several saturated fatty acids (SFA), in particular, myristic (C14:0), palmitic (C16:0) and stearic (C18:0). Micro-RNAs (miRNAs) are a class of small non-coding RNAs (microRNAs), around 22 nucleotides in length, which are contained in exosomes, extracellular membranous organelles with pleiotropic biological functions [4]. Exosomes are formed by the inward budding of multivesicular bodies (MVBs) and are released from the cell into the microenvironment following the fusion of MVBs with the plasma membrane. They are contained in body fluids such as saliva, urine, plasma and milk, and they mediate the delivery of miRNAs to target cells. The double-layer lipid membrane makes functional and exceptionally stable exosomes both in human and cow milk. Indeed, milk-derived miRNAs are resistant to high temperature, low pH, multiple freeze-thaw cycles and RNase treatment [5]. The remarkable stability of the endogenous milk-derived miRNAs implies that infants can intake miRNAs along the lactation according to Gu et al. [6]. The effects of such RNA molecules on health of young and adult milk consumers represent a relevant aspect to explore. Bovine miRNAs are produced by mammary cells and are involved in milk fat metabolism along the lactation [7]; according to Lin et al. [8,9] the presence of some miRNAs, particularly 103 and 27a, in goats’ mammary cells affects the expression of the *SCD* gene. Several factors influence the expression of genes involved in fat metabolism; within them, diet amino acids, lipids and vitamins modulate the miRNA transcriptome of mammary cells in goat milk [10]. In addition, some studies indicated that dietary polyunsaturated fatty acids (PUFAs) might affect the miRNA profile in human [11] and rats [12,13]. On this basis, the aim of this trial was to study the influence of pasture, which is known to be rich in PUFAs, particularly α-linolenic and linoleic acids, on both the *SCD* gene and miRNA 103 expression in goat milk. To date no data are available concerning the possible effects of pasture on miRNA expression in milk.

## 2. Materials and Methods

### 2.1. Animals, Diets and Management

The trial was performed according to the Animal Welfare and Good Clinical Practice (Directive 2010/63/EU) and was approved by the local Bioethics Committee (prot. number PG/2019/0070005). In a farm located at Casaletto Spartano, Salerno province, Southern Italy, at 832 m.s.l., 16 Cilentana dairy goats (3rd parity; 50.5 ± 1.8 kg body weight), delivered in February, were divided into two groups (G, grazing, and S, stable) homogeneous for milk yield at the previous lactation (1590 ± 112 g/h/day).

Group G had free access to pasture (9:00 am to 4:00 pm), constituted by 60% Leguminosae (*Trifolium alexandrinum*, and *Vicia* spp.) and 40% Graminee (*Bromus catharticus, Festuca arundinacea* and *Lolium perenne*) and received a supplement of 500 g/head concentrate. In order to ensure similar protein intake, group S in addition to the same administration of concentrate received alfalfa hay (CP/DM 164 g/kg; NDF/DM 432 g/kg and UFL/kg DM 0.75). Indeed, in our previous trials carried out in the same area [14,15] the protein content of pasture was close to 16% dry matter (DM). The hay daily ingestion, after measuring of the refusals, was 1.1 kg/head.

Samples of pasture were monthly collected from three different areas (2.5 m^2^ each) at no less than 3 cm from the ground. After weighing, samples were air-oven dried at 65 °C, milled through a 1 mm screen and stored.

The chemical composition of feeds was analyzed according to Van Soest et al. [16] and Official Methods of Analysis (AOAC) [17] while the net energy was calculated as suggested by Alimentation Des Ruminants (INRA) [18]. 

Fatty acid (FA) profiles of pasture, alfalfa hay and concentrate were determined using the total fat extraction [19]; FA transmethylation [20,21]; FA quantification by gas chromatograph (ThermoQuest 8000 TOP gas chromatograph, equipped with flame ionization detector; ThermoElectron Corporation, Rodano, Milan, Italy) set according to Tudisco et al. [22] and FA peaks identification by comparing them with those of the standard mixture (Larodan Fine Chemicals, AB, Limhamnsgardens Malmo, Sweden).

In Table 1, chemical composition, nutritive value and fatty acids profile of feeds are reported.

### 2.2. Milk Analysis

Milk was completely suckled by kids up to day 60; therefore, yield was recorded every day and representative samples (weighted average of the two daily milking), were analyzed monthly, from April to August, for protein and fat concentration by Milkoscan 133B (Foss Matic, Hillerod, Denmark). Total fat was obtained by a mixture of hexane/isopropane (3/2 v/v) [23].

The FA profile was determined using the same procedures for feeds with additional standards for CLA isomers (Larodan Fine Chemicals, AB, Limhamnsgardens Malmo, Sweden).

According to Lock and Garnsworthy [24] the C14:1/C14:0 ratio was used to represent the *SCD* activity index.

Milk samples were centrifuged in order to obtain the milk somatic cell pellet. The total RNA extraction and the SCD gene expression were determined as described by Tudisco et al. [25]. Complementary DNA (cDNA) samples obtained by Quantiscript Reverse transcriptase (Qiagen) were amplified using RT-PCR (ABI Prism 7300 System, Applied Biosystems, Foster City, California, USA). The threshold cycle numbers (Ct) of the SCD mRNA were normalized using the mean Ct of housekeeping genes following the formula: 2^−(CT^_SCD_
^− CT^_housekeeping_^)^.

The expression of miRNA 103 was evaluated in whole milk, briefly, a 200 µL/sample of milk was purified using mirVana miRNA isolation kit (Ambion, USA) following the manufacturer’s instructions. The quantity and quality of purified total RNA were determined as described by Tudisco et al. [25] for *SCD* expression. Total RNAs were stored at −80 °C for further use. For amplification of miRNA, DNase I-digested total RNA was polyadenylated and reverse transcribed using a Mir-X™ miRNA First Strand Synthesis Kit (Clontech Laboratories, Inc., CA, USA) to prepare cDNAs, and subjected to quantitative real-time PCR using a SYBR Advantage qPCR Premix (Clontech Laboratories, Inc., CA, USA) with the provided miRNA reference gene (U6). The qRT-PCR analysis was performed in triplicate on ABI Prism 7300 System (AppliedBiosystems, Foster City, California, USA). The 25 μL PCR contained 2.0 μL of the RT product (template), 12.5 μL 2× SYBR Advantage Premix, 9 μL ddH_2_O, 0.5 μL 50× ROX Dye, 0.5 μL miRNA-specific primer (10 μM) and 0.5 μL mRQ 3′primer. The default thermal profile used for PCR amplification consisted of 95 °C for 10 min, followed by 40 cycles of 95 °C for 15 sec and 60 °C for 60 s. The same conditions were performed on an equal amount of RNase-free water as a negative control. The miR-103-specific primer (3’-AGCAGCATTGTACAGGGCTA) was synthesized by Eurofins Genomics (Eurofins Genomics S.r.l., Milano, Italy). Ct values determined for each sample were normalized against the values for U6. The relative fold change in expression to U6 was calculated using 2^−(Ct^_103miRNA_
^− Ct^_U6_^)^.

### 2.3. Statistical Analysis

Data were analyzed using the MIXED procedure of the JMP® (Version 9 SW, SAS Institute Inc., Cary, NC, USA, 1989–2019) [26] for repeated measures over time. The goat was considered as the experimental unit.

The following model was used:Y_ijk_ = µ + DT*_i_* + G*_(i)j_* + ST*_k_* + (DT + ST)*_ik_* + Ɛ_ijk_, where Y_ijk_ = mean of the response variable, µ = population mean, DT*_i_* = effect of the dietary treatment (i = 2; S and G), G*_(i)j_*= random effect of goat within the treatments, ST*_k_* = effect of sampling time (k = 5; April, May, June, July, August), (DT × ST)*_ik_* = fixed effect of interaction between dietary treatment and sampling time and ε_ijk_ = experimental error.

The comparison among the mean values was performed by using the Tukey’s test. Differences were considered statistically significant at *p* < 0.05.

## 3. Results

No refusals were found. Body weights (BW) were not different between the groups along the trial.

Milk yield was unaffected by dietary treatment while group G showed significantly higher fat (4.10% vs. 2.94%; *p* < 0.01) and protein percentage (3.43% vs. 3.25%; *p* < 0.01) than group S (Table 2). Concerning milk fat (Figure 1), the differences between groups were more evident in May and June; in July both groups showed a decrease of values, which was more severe in group G.

As depicted in Table 3, group S showed significantly higher levels of SFA (74.86% vs. 71.42%; *p* < 0.01) as well as of the C14:1 cis9/C14:0 ratio (0.018% vs. 0.009%; *p* < 0.05), and lower values of monounsaturated fatty acids (MUFA, 21.76% vs. 24.39%; *p* < 0.01). Total polyunsaturated fatty acids (PUFA, 3.38% vs. 4.19%; *p* < 0.01), α-linolenic acid (C18:3, 1.35% vs. 0.80%; *p* < 0.01) and total omega 3 (1.48% vs. 0.89%; *p* < 0.01) were significantly higher in group G. In such group, total CLA were twice than in group S (0.646% vs. 0.311%; *p* < 0.01) mainly because of the differences in cis9 trans 11 CLA (0.623% vs. 0.304%; *p* < 0.01).

Milk cis9 trans 11 CLA in grazing group was significantly (*p* < 0.01) higher in August (Table 4) according to the highest value of both linoleic and α-linolenic acids in the pasture (Table 5)

The RNA yield extracted from 300 mL of milk was 4.05 ± 0.9 μg and 3.92 ± 0.6 μg for groups S and G, respectively. These values are considered sufficient for the following analysis [27]. The purity of extracted RNA was measured by a UV spectrophotometer (A260/A280 ratio was 1.8; A260/A230 ratio was 2.1 for all the samples). The mean RNA Integrity Number (RIN) obtained for all extracted samples was 7.7 ± 0.5 and 7.9 ± 0.2, for groups S and G, respectively, thus higher than 6, the threshold to define the quality of RNA [28].

The results of the PCR real time analysis showed that *SCD* expression (AU: Arbitrary unit) was higher, although not significant, in grazing animals either as mean value (AU: 0.703 vs. 0.589) or at each sampling (Figure 2). In this group, *SCD* expression decreased from April to June, increased in July and decreased again in August, while it was almost unvaried along the trial in group S.

The mean yield of miRNA 103 extracted from 200 µL of milk was μg 19.2 ± 1.4 and 18.6 ± 1.1, for group S and G, respectively. The expression of miRNA 103 was higher, although not significant, in group G (AU: 0.417 vs. 0.390) with a similar trend for both groups: Decreasing from April to June, increasing in July and falling down in August (Figure 3).

## 4. Discussion

Milk yield was not different between groups while grazing goats showed significantly higher levels of milk fat. This was probably due to the higher intake of structural carbohydrates; indeed, pasture neutral detergent fiber (NDF) was higher than that of alfalfa hay (see Table 1). The structural carbohydrates are fermented by the rumen cellulolytic bacteria with production of acetic acid, precursor of short and a large part of the medium chain milk fatty acids. Milk fat was similar between groups in July, when grazing group showed the lowest value, which, according to Loewenstein et al. [28] could be due to a depression of pasture quality by the high temperatures.

Anyway, energy requirements were satisfied for both groups. As reported by Rubino [29] the medium pasture ingestion of the local genotype goats is equal to 20 g DM/kg BW while energy needing for maintenance and milk synthesis are 0.0365 UFL/kg metabolic weight (MW = BW^0.75^) and 0.41 UFL/kg fat-corrected milk (4% fat), respectively. In the present trial, the goats from the grazing group weighed 50 kg BW and the intake was 1 kg DM of pasture, equal to 0.76 UFL (see Table 1). In addition, they yielded 1.793 kg milk with 4.1% fat thus their energy requirement was 1.42 UFL (0.69 UFL maintenance, plus 0.73 UFL milk synthesis). Energy deficiency was covered by the concentrate (1.03 UFL/kg DM). Similarly, group S met energy requirements by intake of 1.1 kg DM of alfalfa hay (0.75 UFL/kg DM) plus the concentrate.

Concerning milk fatty acid profile, α-linolenic acid was significantly higher in group G, probably due to the higher level of this acid in the pasture compared to the alfalfa hay (42.6% vs. 38.8%, see Table 1). This result agrees with that reported by Shroeder et al. [30] who found a significant increase of α-linolenic acid in milk of grazing cows compared to cows fed total mixed ration (g/100 g 0.57 vs. 0.07).

Milk CLA was significantly higher in the grazing group, according to the results reported by Dhiman et al. [31] in cows, Nudda et al. [32] in sheep and goats, D’Urso et al. [14], Tudisco et al. [15,22,33] and Zicarelli et al. [34] in goats. Bergamo et al. [35] and Secchiari et al. [36] found higher levels of α-linolenic acid in milk from buffaloes housed in a stable but fed fresh forage rather than a total mixed ration. In contrast, Jahreis et al. [37] did not observe differences in milk CLA between grazing and stabled goats.

Noteworthy, milk CLA in grazing group was significantly (*p* < 0.01) higher in August (see Table 4) according to the highest value of both linoleic and α-linolenic acids in the pasture (see Table 5), as reported by other authors [38,39,40].

On the contrary, pasture linoleic and α-linolenic acids did not similarly influence *SCD* expression. Indeed, in the grazing group, *SCD* expression decreased from April to June, increased in July and decreased again in August (see Figure 2), showing an opposite trend to the pasture linoleic acid, while α-linolenic acid did not change along the experiment, except for an increase in August. These results could demonstrate a higher influence of omega 6 PUFA than omega 3 PUFA on the *SCD* expression.

*SCD* activity is measured comparing the ratio product/substrate of some fatty acids. As described by Lock and Garnsworthy [24], the best marker of *SCD* activity is the c9C14:1/C14:0 ratio since all the C14:0 in milk fat comes from the synthesis in the mammary gland; as a consequence, desaturation is the only source of C14:1. In the present trial, the values of *SCD* activity showed an opposite trend compared to that of the expression of the gene encoding for the same enzyme. Accordingly, the supplementation of sunflower seed oil [41] and linseed oil [42] to maize silage-based diets for goats did not influence *SCD* expression and activity, a similar supplement to a grass hay-based diet only reduced the *SCD* activity [43]. Bernard et al. [44] observed a similar phenomenon supplementing soya beans to lucerne hay-based diets. Bernard et al. [45] underlined the importance of the interaction among ingredients of diet taking into account that rumen-bypass PUFA or bio-hydrogenation intermediates can inhibit *SCD* activity via transcriptional or post-transcriptional mechanisms.

The trend of miRNA 103 expression was similar between the groups, thus no effect of dietary treatment was revealed. Indeed, in both groups, the miRNA 103 decreased from April to June, increased in July and decreased again in August. The miRNA expression was affected only by the stage of lactation according to Avril-Sassen et al. [46], Chen et al., [47] and Wang et al. [48] who found a different expression of miRNAs in the mammary gland along the lactation in mouse and cow and mainly to Dong et al. [49] who reported the lowest expression level of miRNAs at the peak of lactation in goats.

In addition miRNA and SCD gene expression showed similar trends as reported also by Lin et al. [8] who found a high correlation between the overexpression of miRNA103 in mammary cells and the increased expression of genes involved in fat synthesis with accumulation of triglycerides and of a part of unsaturated fatty acids in milk. Indeed, fat synthesis in the mammary epithelial cells recognizes several metabolic processes: An initial neo-synthesis and a subsequent desaturation by *SCD* with conversion to triglycerides [50,51]. Importantly, further larger scale studies are in due course to confirm these preliminary results. In particular, to obtain a number of replicates that should allow a statistical analysis based on groups and to assess other aspects such as the relation between somatic cell count and miRNA expression as shown by Mura et al. [52]. It would be also interesting to investigate, in subsequent studies, how different types of pasture, characterized by different forage essences, can influence milk characteristics. Indeed, to our knowledge this is the first observation of the effects of pasture on miRNA expression in milk from ruminant species.

## 5. Conclusions

The pasture significantly affected milk fat as well as the fatty acid profile in goats, in particular CLAs showed higher levels in milk from grazing animals with beneficial effects for human health. The dietary treatment affected only the *SCD* trend, while that of miRNA 103 appeared influenced only by the stage of lactation. Due to the increasing interest of consumers for the healthy aspects of foods of animal origin, an important debate is in due course in the scientific community concerning those factors affecting milk quality in terms of human health. Thus, the results of this study add useful information that increases the knowledge on this topic.

## Figures and Tables

**Figure 1 animals-09-00606-f001:**
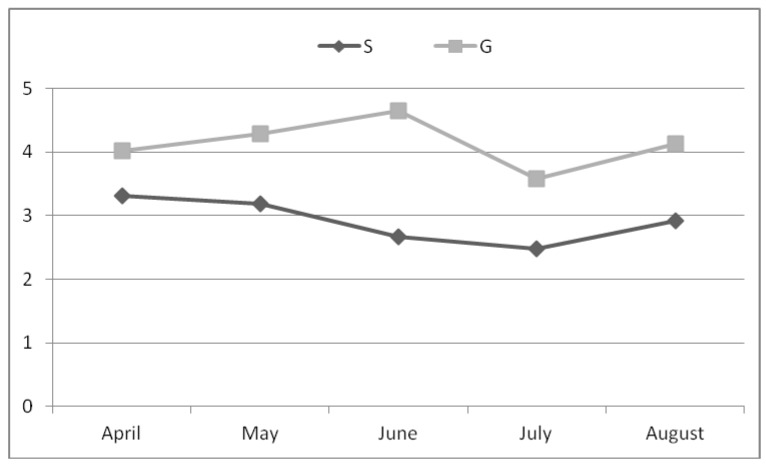
Milk fat (%) along the trial.

**Figure 2 animals-09-00606-f002:**
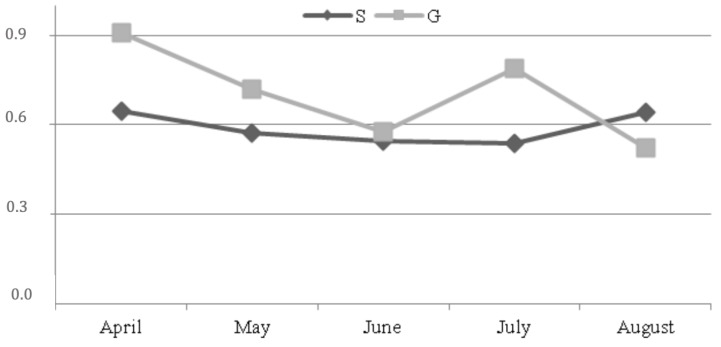
*SCD* expression (AU: Arbitrary unity) along the trial (*SEM = 0.362). * standard error of mean.

**Figure 3 animals-09-00606-f003:**
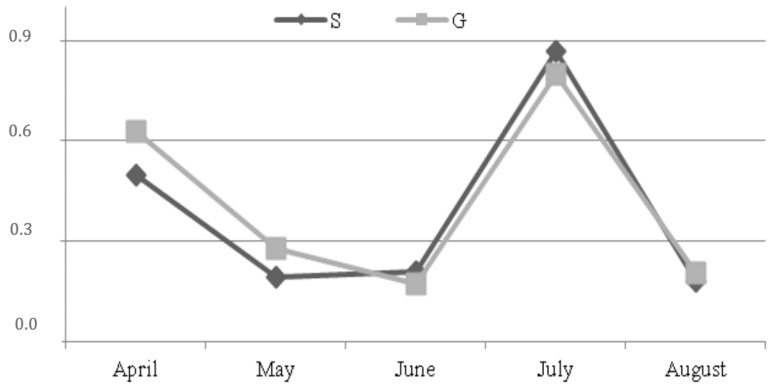
Expression of miRNA 103 (AU: Arbitrary unit) along the trial (*SEM = 0.121). * standard error of mean.

**Table 1 animals-09-00606-t001:** Chemical composition, nutritive value and fatty acid profile of feeds.

Chemical Composition (g/kg DM)	Hay	Concentrate *	Pasture
Crude protein	164.2	180.0	167.0
Ether extract	16.3	30.0	19.0
NDF	434.1	270.0	495.0
ADF	311.4	115.0	345.0
ADL	54.3	30.0	50.0
UFL (kg DM)	0.75	1.03	0.76
Fatty acid profile (% of total FA)			
SFA	23.0	24.6	17.6
MUFA	8.4	16.0	6.4
PUFA	68.6	59.4	76.0
C18:2	18.1	46.0	26.0
C18:3	38.8	9.3	42.6

DM, dry matter; NDF, neutral detergent fiber; ADF, acid detergent fiber; ADL, acid detergent lignin; UFL, unit feed for lactation; SFA, saturated fatty acid; MUFA, mono-unsaturated fatty acid; PUFA, polyunsaturated fatty acid. *Ingredients (g/kg DM): Soft wheat bran 300; soybean solvent extract 130; corn 130; sunflower 105; citrus pulp 80; dried beet pulp 79; corn gluten feed 70; sugarcane molasses 77; CaCO_3_ 15; dicalcium phosphate 7; vitamin-mineral premix 2 and NaCl 7.

**Table 2 animals-09-00606-t002:** Milk yield (g/head/day) and chemical composition (%).

Milk	Yield		Fat		Protein		Lactose	
Group	S	G	S	G	S	G	S	G
	1798.1	1793.0	2.94	4.10	3.25	3.43	4.02	3.99
Group effect	NS		**		**		NS	
Sampling effect	**		**		NS		**	
G ^×^ S	NS		NS		NS		NS	
SEM	167.5		0.536		0.336		0.220	

**, *p* < 0.01; NS, not significant. SEM, standard error of mean.

**Table 3 animals-09-00606-t003:** Milk fatty acid profile (% total fatty acids).

Milk Fatty Acids Profile	S	G	Group Effect	Sampling Effect	*G ^×^ S*	SEM
C4:0	0.0002	0.0002	NS	NS	NS	0.001
C6:0	0.039	0.02	NS	**	NS	0.073
C8:0	0.949	0.883	*	**	*	0.577
C10:0	8.96	10.08	**	**	*	2.046
C12:0	5.41	3.99	**	*	**	0.901
C14:0	10.56	13.00	**	**	NS	1.025
C14:1 cis9	0.199	0.119	NS	**	*	0.111
C16:0	34.43	30.22	**	**	**	2.350
C16:1	0.82	0.50	**	**	**	0.190
C17:0	0.742	0.761	NS	NS	NS	0.236
C17:1	0.233	0.213	NS	**	NS	0.093
C18:0	13.39	12.01	**	**	**	1.606
Total C18:1 trans	0.678	2.007	**	NS	**	0.374
Total C18:1 cis	19.68	21.48	NS	NS	NS	2.344
C18:2 omega-6	2.048	1.906	NS	**	NS	0.374
C18:3 omega-3	0.797	1.351	**	*	**	0.251
C20:0	0.187	0.213	NS	NS	*	0.11
C20:1	0.126	0.043	NS	NS	NS	0.238
C21:0	0.046	0.054	NS	NS	**	0.021
C20:2	0.008	0.014	**	*	NS	0.011
C22:0	0.076	0.108	*	**	**	0.060
C20:3 omega-6	0.018	0.051	*	NS	NS	0.048
C22:1	0.016	0.020	NS	*	NS	0.011
C20:3 omega-3	0.032	0.050	NS	NS	NS	0.056
C20:4 omega-6	0.079	0.072	NS	NS	NS	0.064
C23:0	0.035	0.026	NS	**	*	0.036
C22:2	0.022	0.018	NS	**	NS	0.016
C24:0	0.036	0.056	**	*	**	0.025
C20:5 omega-3	0.040	0.053	**	NS	NS	0.022
C24:1	0.006	0.009	NS	**	NS	0.011
C22:6 omega-3	0.024	0.031	NS	**	*	0.018
cis-9 trans-11 CLA	0.304	0.623	**	**	*	0.145
trans-10 cis-12 CLA	0.008	0.023	**	NS	NS	0.018
SFA	74.860	71.420	**	**	**	3.008
MUFA	21.760	24.388	**	**	**	2.64
PUFA	3.380	4.192	**	NS	*	1.008
∑ CLA	0.312	0.646	**	NS	**	0.153
∑ omega-3	0.893	1.485	**	*	**	0.160
∑ omega-6	2.145	2.029	NS	**	*	1.032
C14:1 cis9/C14:0	0.018	0.009	*	*	NS	<0.01

CLA, conjugated linoleic acid; SFA, saturated fatty acids; MUFA, monounsaturated fatty acids; PUFA, polyunsaturated fatty acids. *, *p* < 0.05; **, *p* < 0.01; NS, not significant. SEM, standard error of mean.

**Table 4 animals-09-00606-t004:** Milk fatty acid profile (% of total fatty acid) along the trial.

Group	April		May		June		July		August		Group Effect	Sampling Effect	*G ^×^ S*	SEM
	S	G	S	G	S	G	S	G	S	G				
C14:0	10.7	11.1	10.2	11.7	10.3	14.5	11.1	14.5	10.5	13.3	**	**	NS	1.025
C14:1	0.18	0.07	0.12	0.17	0.33	0.21	0.16	0.08	0.21	0.07	NS	**	*	0.111
C18:2	2.35	2.52	2.29	2.64	2.04	1.03	1.68	1.67	1.88	1.67	NS	**	NS	0.374
C18:3	1.16	1.09	0.89	1.30	0.58	1.24	0.74	1.68	0.61	1.45	**	*	**	0.251
c9 t11 CLA	0.48	0.61	0.30	0.55	0.29	0.52	0.21	0.68	0.24	0.75	**	**	*	0.145
C14:1/C14	0.015	0.006	0.010	0.016	0.022	0.020	0.011	0.007	0.015	0.006	*	*	NS	< 0.01

*, *p* < 0.05; **, *p* < 0.01; NS, not significant. SEM, standard error of mean.

**Table 5 animals-09-00606-t005:** Fatty acid profile (% of total fatty acids) of pasture as function of sampling months.

Pasture Fatty Acid Profile	April	May	June	July	August
SFA	22.8	16.2	16.2	20.9	11.8
MUFA	8.9	5.0	5.0	5.9	7.4
PUFA	68.3	78.8	78.8	73.2	80.8
C18:2	20.0	28.2	34.2	19.2	28.2
C18.3	32.6	42.6	44.4	40.6	48.2

SFA, saturated fatty acids; MUFA, mono-unsaturated fatty acids; PUFA, polyunsaturated fatty acids.

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
