# Peer review of "Influence of Pasture on Stearoyl-CoA Desaturase and miRNA 103 Expression in Goat Milk: Preliminary Results"

_animals, 2019, doi:10.3390/ani9090606_

Round 1

Reviewer 1 Report

The paper entitled "Influence of pasture on Stearoyl-CoA desaturase and miRNA 103 expression in goat milk" deals with an interesting topic, of relevant interest for the scientific community.

Some results are highly known, but data on miRNA are innovative and interesting.

The draft presents, according to me, a unique weak point, related to the soundness of the statistical analysis.

As authors declared in the paper, the statistical unit is the goat, although the animals are fed as group. I understand the difficulties of performing on farm studies with individual feeding, above all, on pasture, but from a scientific point of view, the results are affected by this aspect. Really, the statistical unit is the group, and so, there are not sufficient replications for applying a statistical analysis.

So, I suggest to authors to add at the end of the title ": preliminary results", and discuss this aspect in the "Discussion" section, stating that the results obtained are the first according their own best knowledge, but these results have to be verified in larger scale studies. Besides authors should highlight that their results have been obtained on THAT pasture, and that further researches should investigate on which kind of pasture characteristics are necessary for achieving the results of improved milk quality.

Finally, please verify the numeration of the papers cited in the "References" section. There is a shift in references due to the voices 4. and 5. empty.

Author Response

Dear Sir,

attached you can find the answers to your questions. Thank you for your suggestions that have improved the paper.

Best regards

Reviewer 2 Report

Dear Author,

the manuscript addresses an interesting topic within the field of dairy production: the influence of feeding on milk composition and nutritional characteristics, through the regulation of genes expression. In particular, the effect of pasture on the expression of both a gene and a miRNA involved in biosynthesis of CLAs was evaluated in dairy goats.

The manuscript is sound, but some improvements are requested:

- the simple summary should be made more effective, in particular the sentence “These are two…” is vague and not enough justified ;

- the references  should be revised thoroughly (references 4 and 5 are missing, and the numbering from 6 on is incorrect);

- the expression of SDC was determined on RNA extracted from milk somatic cell pellet. Actually, somatic cells include both epithelial cells from the mammary gland  and immune cells from blood. It would be interesting to know the trend of somatic cell count during the trial;

- in my opinion, it should be highlighted that miRNA 103 expression was evaluated not in exosomes but in whole milk.

Minors remarks:

Line 20: effect on…

Line 23: there is an important ongoing debate  in the scientific …

Line 55: product, obtained during…

Line 57: Micro-RNAs (miRNAs)…

Line 69-70:  along the lactation. According to … and check the references.

Line 76: in goat milk. To date no data…

Line 97 Fatty acid (FA) profiles of pasture, alfalfa hay and concentrate were determined using the following methods: total fat…

Line 114-114 representative samples (weighted average of the  two daily milking) were analyzed monthly, from April to August, for …

Line 121: The total RNA was extracted from milk somatic cell pellet and the SDC gene expression was determined as described by…

Line 128: above ?

Line 151: test. Differences …

Line 260: could demonstrate or seems demonstrate

Line 265: is the only source …

Line 281: in addition, miRNA and SDC gene expression showed similar trends as reported…

Author Response

(The authors gave the same response as above.)

Round 2

Reviewer 1 Report

According to my opinion the paper is worthy of publication in its present form. the authors recognized clearly the weak points of the paper and honestly declare them in discussion section, opening to further researches